# Urinary Levels of miR-491-5p and miR-592 as Potential Diagnostic Biomarkers in Female Aging Patients with OAB: A Pilot Study

**DOI:** 10.3390/metabo12090820

**Published:** 2022-08-31

**Authors:** Philippe G. Cammisotto, Abubakr H. Mossa, Samer Shamout, Lysanne Campeau

**Affiliations:** 1Lady Davis Institute, McGill University, Montreal, QC H3T 1E2, Canada; 2Division of Urology, Department of Surgery, Jewish General Hospital, McGill University, Montreal, QC H3T 1E2, Canada

**Keywords:** miRNAs, NGF, matrix metalloproteinase-9, p75^NTR^, urine, OAB, LUTS

## Abstract

Women with overactive bladder syndrome (OAB) have a lower urinary ratio of nerve growth factor (NGF) to its precursor (proNGF) compared to healthy controls. MicroRNAs related to NGF and proNGF metabolism and to their receptors may be present in urine and may possess diagnostic value. Urine and blood samples from 20 control and 20 OAB women (50–80 years) were obtained, together with validated questionnaires and other clinical parameters. The relative expression of urinary microRNAs was measured with RT-qPCR. MiR-491-5p, which negatively controls the translation of the matrix metalloproteinase-9 (MMP-9), the main enzyme degrading NGF, was significantly decreased in OAB. Similarly, miR-592, which represses p75^NTR^ receptor synthesis, was down-regulated in OAB. Age, renal function and insulin resistance did not affect these results. ROC curves confirmed the high sensitivity of miR-491-5p and miR-592 for diagnosis. On the other hand, miRNAs involved in the expression of proNGF, of survival receptor TrkA and of markers of nerve integrity were similar between groups. The detection of miR-491-5p and miR-592 in urine could be a useful and non-invasive tool for the diagnosis of OAB syndrome in aging women.

## 1. Introduction

OAB is defined by the International Continence Society (ICS) as “urinary urgency, usually accompanied by increased daytime frequency and/or nocturia, with urinary incontinence (OAB-wet) or without (OAB-dry), in the absence of urinary tract infection or other detectable disease” [1]. The EPIC multicenter population-based study estimated an overall OAB prevalence of 11.8% (10.8% in men versus 12.8% in women), with an exponential increase in prevalence with advancing age. Women aged 40–59 years are more likely to experience storage symptoms compared to men (56% versus 51%) [1]. OAB affecting preferentially women has been linked to a series of factors, including advanced age and menopause, an increase in body mass index (BMI), and high parity rates [2]. Nevertheless, despite years of ongoing research, OAB diagnosis as a syndrome remains purely symptomatic and the phenotyping of potential different subtypes has been unsuccessful [3]. Nevertheless, it is now widely accepted that the central and peripheral nervous systems are involved in its pathophysiology [4].

Neurotrophins are a family of growth factors essential for the development, survival and synaptic remodeling of neurons [5]. Among them, nerve growth factor (NGF) and its precursor (proNGF) activate numerous intracellular pathways by binding cell surface receptors, namely, tyrosine kinase receptor A (TrkA) and p75 neurotrophin receptor (p75^NTR^) [5]. In particular, NGF has a high affinity for TrkA and promotes nerve growth and cell survival, while proNGF preferentially binds p75^NTR^ to trigger inflammatory pathways, neurodegenerative processes and apoptosis [6]. However, the global outcome of these interactions is complex to predict, as it depends on the proNGF/NGF ratio and the relative amounts of TrkA versus p75^NTR^ [7]. The proNGF/NGF imbalance is indeed related to different neurological and urological pathologies [8,9].

Regardless, many studies have focused on the urinary level of NGF as a potential biomarker for OAB [10]. Other clinical data suggest a link between levels of NGF in bladder tissue and OAB, among other bladder pathologies [11]. More recently, we reported that in the urine from an aging female population, the ratio proNGF/NGF rather than NGF/creatinine could represent a more reliable trait of OAB [12]. We also found that the protein convertase levels for MMP-9, the main enzyme responsible for NGF proteolysis, were increased, while the synthesis of proNGF was not affected [12].

Other markers for diseases that provide valuable clinical or diagnostic information for neurodegenerative diseases in particular are microRNAs. They are abundant, remarkably stable in fluids, including plasma and urine, and have the advantage of being able to undergo multiple thaw–freeze cycles without it affecting their concentration or the quality of the qPCR products [13]. Recently discovered as short, noncoding RNAs, they are involved in the post-transcriptional regulation of messenger RNAs (mRNAs), and they appear to be potentially disease-specific [14]. Stress response and apoptosis are among the many processes affected by microRNAs.

In order to further understand the mechanisms involved in OAB from the perspective of a dysregulation in NGF processing and signaling, we analyzed selected miRNAs implicated in the synthesis of proNGF (miR-98-5p, let-7b-5p and let-7d-5p), MMP-9 (miR-491-5p and miR-885-5p), TrkA (miR-221-5p and miR-92a-3p) and p75^NTR^ (miR-592) receptors and in the nervous system health (miR-21-5p, miR-132 and miR-212). We compared their levels in the urine from women with or without OAB in an aging population.

## 2. Materials and Methods

### 2.1. Patient Samples

OAB subjects, women between the age of 50 and 80 years old who presented with a clinical diagnosis for OAB (with or without treatment) at a urology practice at the Jewish General Hospital, Montreal, Canada, were compared to 20 control subjects from the same age group. Eligible participants enrolled in the OAB group (n = 20) were women aged 50 to 80 years, who presented with complaints of OAB symptoms including urinary frequency and urgency, or urge incontinence, for at least 3 months and had not been on any anticholinergic treatment for at least 3 weeks, with a negative screening urine culture (to exclude UTI). Regarding the control group (n = 20), we recruited age-matched normal volunteers or patients attending the same clinic (50–80 years old) who had no urinary symptoms, no current or prior use of OAB medications and a negative urine test for any infection. The exclusion criteria included established diabetes mellitus, a history of malignancies or pelvic radiotherapy, pelvic organ prolapse, urinary tract infection, neurogenic lower urinary tract dysfunction and hepatic or renal impairment (creatinine clearance <70 mL/min). Informed consent was obtained from all patients. The study was approved by the Medical-Biomedical Research Ethics Committee (REC) of the Integrated Health and Social Services University Network for West-Central Montreal (IRB: 2016-328, 15-022, approved on 20 June 2017).

The sample size calculation followed the original project estimation based on the Human Metabolome Data Base. The urine succinate level difference between the control and the disease conditions was 3.4 μmol/mmol creatinine (normal urine succinate of 5.6 and abnormal of 9.0 umol/mmol creatinine), with a standard deviation (sd) of 3.8, study power at 80% and significance at 0.05 [15,16,17].

### 2.2. Demographic and Clinical Differences

Every participant underwent a complete medical history, physical examination, screening urinalysis, 1 day voiding diary and validated symptom questionnaires (see flow diagram, Appendix A). The Voiding diary was used to estimate: 24 h, daytime and nighttime frequencies, total 24 h voiding volume, nocturnal voiding volume, the mean voided volume per micturition and the maximum voided volume. An Overactive Bladder Symptom Score (OABSS), the International Consultation on Incontinence Questionnaire-Short Form (ICIQ-SF) and the Incontinence Impact Questionnaire (IIQ-7) were completed by all participants. Fasting glucose and insulin levels were used to calculate the Homeostatic Model Assessment of Insulin resistance (HOMA-IR) as an indicator for insulin resistance. Significant insulin resistance was indicated by values above 2.9.

### 2.3. Sample Collection and Preparation

Midstream, early morning urine samples were collected by patients in two sterile plastic containers, one kept at 4 °C for urine culture and the other container placed at −20 °C. No dietary restrictions during urine collection were applied. Upon reception, the samples were aliquoted and stored at −80 °C. The laboratory staff was blinded to which samples were OAB samples or controls.

### 2.4. Isolation of miRNAs from Urine

Urine aliquots kept at −80 °C were thawed on ice. Centrifugation was carried out at 10,000 rpm for 20 min to remove cells and cellular debris. Subsequently, miRNAs were isolated from supernatants using a urine miRNA purification kit (Norgen Biotek protocol Corp, Thorold, ON, Canada) according to the manufacturer’s protocol. These columns isolate total miRNAs (cell-free and vesicular/exosomal) as well as small nuclear RNAs [18].

### 2.5. Poly-Adenylation and cDNA Synthesis

Poly(A) tails were added to mature miRNAs using a poly(A) polymerase tailing kit from Lucigen (Middleton, WI, USA). Briefly, purified miRNAs were incubated with ATP (1 mM) and *E. coli* poly(A) polymerase (200 U/mL) for 30 min at 37 °C. The reaction was terminated by placing samples for 5 min at 95 °C. Then, cDNA was synthesized using a custom-made stem loop primer containing a poly-T tail (Integrated DNA Technologies (IDT, Coralville, IA, USA)) and a reverse transcriptase kit (OneScript cDNA synthesis kit) from abm (Richmond, BC, Canada) according to manufacture protocols. RT for the reference gene snU6 was carried out using a specific primer [19]. The total cDNA was quantified using a nanodrop system.

### 2.6. qPCR

All primers were obtained from Integrated DNA Technologies (IDT, Coralville, IA, USA). A universal primer complementary to the stem loop primer was used with forward primers specific to each miRNA of interest (see details on primers and qPCR in Appendix A). The reference gene snU6 had its own set of primers. The qPCRs were carried out using a Sensifast Probe Low-ROX kit containing SYBR-green on an Applied Bioscience 7500 Fast Real-Time PCR under the following conditions: 95 °C for 10 min, then 45 cycles of 95 °C for 15 s and 60 °C for 35 s, followed by a melt curve analysis. Samples were taken in triplicates. Each primer was tested for specificity and efficiency (90–110%). Relative miRNA expressions were analyzed using the 2^−ΔΔCT^ method.

### 2.7. Statistics

Comparisons between groups were performed using a Student t-test (demographics, voiding diary, serum data and questionnaires) or Mann–Whitney test (not normally distributed, miRNAs). Significance was set at *p* < 0.05. One-way ANCOVA was performed to compare the two groups while controlling for confounders (age, HOMA and renal function, which were found to be statistically different between the two groups). Spearman’s correlation was performed between the urinary miRNAs levels and the questionnaires’ scores and voiding diary parameters. A receiver operating characteristic (ROC) was performed to test for sensitivity and specificity. IBM SPSS Statistics ver.23.0 (IBM Co., Armonk, NY, USA) was used.

## 3. Results

### 3.1. Subject Demographics and OAB Symptom Analysis

The OAB group (n = 20) displayed a higher mean age (68.9 ± 11.38) compared to the control subjects (56.25 ± 5.22, *p* < 0.001). There were no significant differences in the demographics and vital signs between the controls and the OAB subjects. Significantly higher HOMA-IR reflecting insulin resistance and eGFR values reflecting worse renal function were found in the OAB group (Table 1). The prevalence of MetS (8 out of 20, 40%) and hypertension (13 out of 20, 65%) was significantly higher in the OAB group compared to the control group (4 out of 20 for both MetS and hypertension together). OAB treatment with anticholinergic drugs or β3-adrenergic agonists was reported in 12 out of the 20 OAB patients.

The significantly higher total questionnaire scores for OAB symptom severity and its impact on the patients’ quality of life in the OAB group conformed with their clinical diagnosis (Table 1). Similarly, higher 24 h frequency and higher nighttime frequency were seen in the OAB group, with a significantly lower maximum voided volume (Table 1).

### 3.2. Quantitative Detection of miRNAs in Urine and Analysis of Data

ProNGF mRNA transcription is under the direct control of several miRNAs, the most prominent being miR-98-5p, let-7b and let-7d [20]. A decrease in their expressions would increase proNGF synthesis and conversely. Here, the urinary levels of these three miRNAs were not different between the control and OAB groups (Table 2, Figure 1). These observations get along well with our previous data where urinary proNGF levels did not show a significant difference between the control and OAB patients [12].

Subsequently, we analyzed two miRNAs involved in the translation of MMP-9, the main protease digesting mature NGF [21]. The MMP-9 mRNA UTR-3′ region is directly bound by miR-491-5p [22]. The miR-885-5p targets the expression of genes upstream of MMP-9, eventually reducing MMP-9 expression [22]. MiR-491-5p levels were decreased by 71% in the OAB group as compared to the control group (*p* = 0.005) while miR-885-5p was not affected (Table 2, Figure 1). These results highlight the complexity of miRNA gene regulation, depending on whether the control is direct or upstream of the target gene. Interestingly, urinary MMP-9 enzymatic activity was reported to be significantly higher in OAB patients compared to the control group [12], an increase that could be explained at least in part by the loss of miR-491-5p post-transcriptional control.

The p75^NTR^ mRNA has a conserved miR-592 binding sequence [23], while TrkA mRNA expression is correlated with miR-221-5p and miR-92a-3p [24,25]. Accordingly, an inverse correlation between miR-592 and p75^NTR^ expression has been demonstrated. We found here that the level of miR-592 in the OAB group was decreased by 48% compared to the controls (*p* = 0.010), suggesting an upregulation of p75^NTR^ (Table 2, Figure 1). On the other hand, miR-221-5p and miR-92a-3p expressions were similar among both groups (Table 2, Figure 1). These results are in favor of an upregulation in the pro-inflammatory environment during OAB. Finally, the expressions of miR-21-5p, miR-132 and miR-212-5p—three miRNAs involved in peripheral and central nervous system health, neuron morphogenesis and neuronal function—displayed no differences between the groups [26,27]. In our sample of the population, the nervous system integrity appears to not be affected during OAB (Table 2, Figure 1).

The effects of confounders (age, insulin resistance and renal function) were subsequently assessed for all microRNAs, as they were found to be statistically different between the two groups. They did not affect our results, except for in the case of miR-92a-3p, which became significantly decreased in OAB when taking into account age and eGFR (Table 3). On the other hand, miR-491-5p and miR-592 urinary levels correlated negatively with OAB symptom questionnaires’ scores, indicating that their downregulation is correlated with the severity of OAB symptoms (Table 4). Moreover, their expression bears acceptable diagnostic value for OAB, as illustrated by the significantly large area under the curve (AUC) being 0.763 (*p* = 0.005) and 0.744 (*p* = 0.01) for miR-491-5p and miR-592, respectively (Figure 2).

## 4. Discussion

The present paper reveals for the first time that the urinary levels for miR-491-5p and miR-592 are reduced in aging female patients with OAB when compared to controls. These decreases negatively correlated with OAB symptom severity, even after correcting for age, insulin resistance and renal function.

The urinary protein ratio for proNGF/NGF was found to be increased in female OAB subjects with stable proNGF levels [12]. The examination of the three miRNAs involved in the control of proNGF expression (miR-98-5p, Let-7b-5p and Let-7d-5p) showed that they were indeed found to be unchanged, suggesting that the synthesis of proNGF is not affected in OAB and that the lower NGF levels result from changes in maturation or degradation. Indeed, the increased MMP-9 activity shown in OAB can lead to lower NGF levels, and miR-491-5p, which directly inhibits MMP-9 by binding to MMP-9 mRNA 3′UTR, was potently down-regulated in the OAB group. Interestingly, a similar mechanism exists in patients suffering from Alzheimer’s disease and Down syndrome [28]. In these pathologies, NGF pools are quickly hydrolyzed by the increased presence of MMP-9. On the other hand, we confirmed our results by taking into account aging. Many reports have measured plasma MMP-9 levels during aging and noted that they slightly decrease or remain stable [29]. Contrary to the results for miR-491-5p, miR-885-5p, which targets genes upstream of MMP-9 and eventually affects its expression, was not affected by OAB. These differences can be explained by the complexity of miRNA gene control; direct targeting of mRNA might exert better control than indirect miRNA with a large scale of regulation.

Our second main finding concerns the p75^NTR^ and TrkA receptors; miR-592 is essential in the inhibition of p75^NTR^ expression. In the context of ischemia, miR-592 drops with concomitant increased p75^NTR^ expression [23]. In the present paper, decreased levels of miR-592 in OAB could lead to a potential increase in p75^NTR^ expression in the bladder and/or kidney epithelium cells during OAB, suggesting an increase in inflammatory processes. In contrast, the two miRNAs involved in TrkA expression showed no change between the control and OAB groups. However, miR-92a-3p became significantly different statistically after correcting for age and eGFR. It must be noted that TrkA and P75^NTR^ are differently regulated during aging in the central nervous system [30]. Receptor TrkA is predominant in young animals, but a switch occurs during aging, resulting in the increased expression of p75^NTR^ to the detriment of TrkA [30,31].

Finally, as OAB could have a neurogenic origin, we examined three miRNAs (miR-21-5p, miR-132 and miR-212-5p) essential for nervous system activity. The miR-21-5p binding site is present in neurotrophin NT3 and is known to enhance neurotrophin signaling and preserve the viability of neurons [32]. MiR-221 regulates negatively both NGF and TrkA at both protein and mRNA levels [24]. MiR-132 targets multiple signaling pathways to influence neuron activity [33]. The three miRNAs were unchanged as well, suggesting that in our cohort of OAB patients, the integrity of the peripheral nervous system was relatively preserved. These observations also suggest that the balance between NGF, proNGF expression and their receptors’ signaling are relevant targets to explain the pathology of OAB, as they are involved in the control of pro-inflammatory as well as neuronal survival pathways.

The limitations of our study are common to all miRNA studies: the choice of reference gene, the method of isolation (centrifugation vs. kits), the body fluid studied or the size of the population, their age and the origin of the miRNAs [34]. For example, as opposed to our results for urine, our analysis of the plasma from OAB patients revealed increases in the systemic levels of let-7b-5p, miR-98-5p and miR-92a-3p, studied as the systemic regulators of beta-3 adrenergic receptors and GEF gene expression [35]. Another difficulty in analyzing miRNAs is that a single miRNA can control up to 100 genes, and conversely, one mRNA can be regulated by dozens of miRNAs. There is also a major difference in gene regulation between miRNAs binding directly to an mRNA of interest and those targeting gene expression upstream of the same gene. Finally, it is difficult to link levels of proteins or miRNAs found in urine to their content in cells or tissues. However, microvesicles, exosomes or microparticles originating from epithelial cells and delivered to biological fluids, are believed to be a mirror of intracellular events [36]. Further studies at the levels of cells and organs will be essential to completing our observations.

## 5. Conclusions

Our study identified a decrease in miR-491-5p and miR-592 in the urine of OAB patients. These results strengthen the hypothesis that an increase in MMP-9 activity could explain the decrease in NGF levels in patients with OAB and further improve our understanding of the mechanisms of OAB. Moreover, the balance between the proapoptotic receptor p75^NTR^ and the survival receptor TrkA appeared to be disrupted, suggesting that neurotrophins could be central players in the pathology of OAB.

## Figures and Tables

**Figure 1 metabolites-12-00820-f001:**
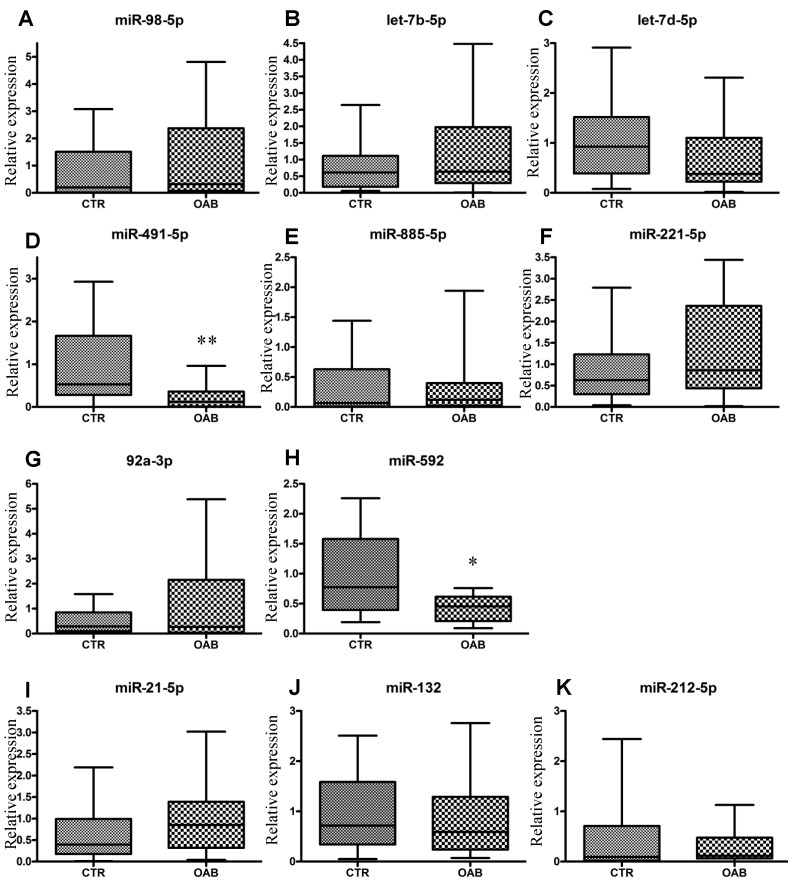
Urinary microRNA expression in control (CTR) and OAB patients. Mean expression levels of urinary miRNA (**A**–**K**) were determined by RT-qPCR using the 2^−ΔΔCT^ method. The midline in the box and whisker plots depicts the median with upper and lower limits representing the minimum and maximum value minus the outliners; * *p* < 0.05, ** *p* < 0.005.

**Figure 2 metabolites-12-00820-f002:**
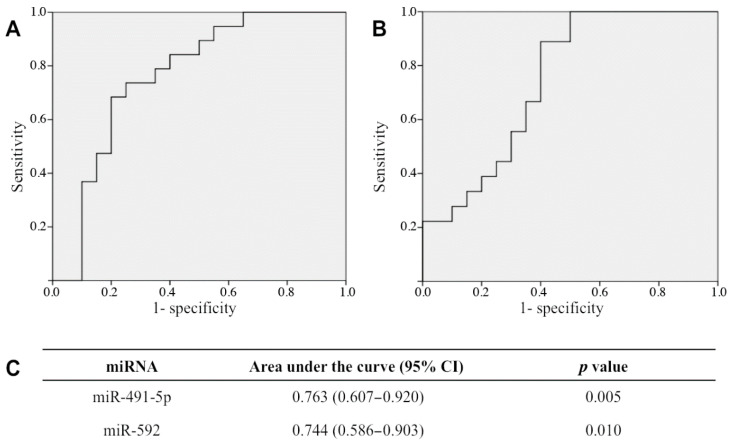
ROC curves for miR-491-5p (**A**) and miR-592 (**B**) were generated from the control (n = 20) and OAB (n = 20) groups. The values for the area under curve (AUC) with a 95% confidence interval (CI) for significant miRNAs are shown in table (**C**).

**Table 1 metabolites-12-00820-t001:** Demographic, serum, symptom questionnaires and urine analyses data compared in the control and OAB groups.

	CTR	OAB Group	*p* Value
Demographic and serum analysis:			
Age (years)	56.25 (5.22)	68.9 (11.38)	<0.001
BMI (kg/m^2^)	29.75 (7.65)	28.82 (5.45)	ns
eGFR (mL/min/1.73 m^2^)	98.5 (14.52)	76 (19.78)	<0.001
HOMA-IR	2.13 (1.03)	3.11 (1.18)	0.020
Total Cholesterol/HDL	3.50 (1.18)	3.23 (0.81)	ns
Questionnaires’ scores:			
OABSS (0–28)	7.3 (3.56)	17.45 (4.45)	<0.001
ICIQ-SF (0–22)	3.26 (3.98)	8.05 (3.83)	<0.001
IIQ-7 (0–100)	2.4 (5.2)	28.9 (23.2)	<0.001
Voiding diary parameters:			
24 h frequency	9.15 (2.28)	11.4 (3.03)	0.012
Daytime frequency	8.5 (2.04)	9.5 (2.09)	ns
Night frequency	0.65 (0.81)	1.9 (1.71)	0.005
24 h voiding volume (mL)	2705 (2346.02)	1859.6 (865.37)	ns
Night voiding volume (mL)	495.25 (253.88)	449.75 (270.77)	ns
Mean voided volume (mL)	322.25 (311.1)	167.36 (75.2)	0.037
Maximum voided volume (mL)	480.75 (193.44)	327.25 (126.7)	0.005

Data are presented as mean (Standard Deviation) for variables compared with an independent *t*-test. Statistically significant differences are reported with a *p* value; (ns) nonsignificant. Abbreviations: Body Mass Index (BMI), estimated Glomerular Filtration Rate (eGFR), Homeostatic Model Assessment for Insulin Resistance (HOMA-IR), Overactive Bladder Symptom Score (OABSS), International Consultation on Incontinence Questionnaire-Short Form (ICIQ-SF) and Incontinence Impact Questionnaire (IIQ-7).

**Table 2 metabolites-12-00820-t002:** miRNA expression (2^−ΔΔCT^) between the control (CTR) and OAB groups without considering confounders.

	CTR	OAB Group	*p* Value
miR-98-5p	0.196 (0.012, 1.508)	0.318 (0.0703, 2.366)	0.343
let-7b-5p	0.608 (0.176, 1.109)	0.633 (0.304, 1.874)	0.584
let-7d-5p	0.922 (0.344, 1.541)	0.379 (0.208, 1.194)	0.130
miR-491-5p	0.534 (0.254, 1.686)	0.118 (0.013, 0.365)	**0.005**
miR-885-5p	0.0633 (0.009, 0.419)	0.124 (0.031, 0.394)	0.327
miR-221-5p	0.626 (0.297, 1.231)	0.859 (0.385, 2.424)	0.299
miR-92a-3p	0.287 (0.089, 0.847)	0.268 (0.032, 2.195)	0.715
miR-592	0.777 (0.375, 1.622)	0.460 (0.234, 0.612)	**0.010**
miR-21-5p	0.398 (0.178, 0.942)	0.854 (0.318, 1.392)	0.224
miR-132	0.713 (0.327, 1.615)	0.593 (0.242, 1.287)	0.384
miR-212-5p	0.089 (0.024, 0.561)	0.113 (0.058, 0.482)	0.756

Data are presented as median (interquartile range, Q1, Q3) for variables compared with the non-parametric Mann–Whitney test. Statistically significant differences are in bold.

**Table 3 metabolites-12-00820-t003:** miRNA expression (2^−ΔΔCT^) between the control (CTR) and OAB groups considering confounders.

	Confounders	CTR	OAB Group	*p* Value
miR-98-5p	Age	0.704 (0.02–1.387)	1.275 (0.592–1.959)	0.283
	HOMA-IR	0.768 (−0.008–1.545)	1.560 (0.753–2.367)	0.179
	eGFR	0.985 (0.332–1.637)	0.994 (0.342–1.647)	0.985
let-7b-5p	Age	0.653 (0.1–1.205)	1.302 (0.731–1.873)	0.142
	HOMA-IR	0.755 (0.209–1.301)	1.262 (0.669–1.855)	0.237
	eGFR	0.779 (0.232–1.325)	1.169 (0.605–1.733)	0.360
let-7d-5p	Age	0.965 (0.593–1.338)	0.726 (0.354–1.099)	0.409
	HOMA-IR	1.161 (0.748–1.575)	0.712 (0.284–1.140)	0.151
	eGFR	0.978 (0.612–1.344)	0.713 (0.347–1.079)	0.346
miR-491-5p	Age	0.999 (0.619–1.378)	0.225 (−0.167–0.616)	**0.013**
	HOMA-IR	0.909 (0.531–1.287)	0.266 (−0.141–0.672)	**0.030**
	eGFR	1.1 (0.735–1.465)	0.118 (−0.258–0.494)	**0.001**
miR-885-5p	Age	0.255 (−0.028–0.538)	0.389 (0.106–0.671)	0.548
	HOMA-IR	0.272 (−0.034–0.578)	0.453 (0.147–0.759)	0.413
	eGFR	0.328 (0.059–0.596)	0.316 (0.047–0.585)	0.954
miR-221-5p	Age	0.744 (0.233–1.256)	1.425 (0.929–1.921)	0.086
	HOMA-IR	0.768 (0.276–1.259)	1.216 (0.725–1.708)	0.216
	eGFR	0.749 (0.249–1.249)	1.421 (0.935–1.906)	0.079
miR-92a-3p	Age	0.276 (−0.589–1.142)	1.668(0.885–2.4510	0.036
	HOMA-IR	0.604 (−0.332–1.540)	1.446 (0.581–2.311)	0.206
	eGFR	0.318 (−0.525–1.161)	1.633 (0.868–2.398)	**0.039**
miR-592	Age	0.981 (0.705–1.257)	0.445 (0.151–0.739)	**0.019**
	HOMA-IR	1.096 (0.796–1.396)	0.405 (0.082–0.728)	**0.005**
	eGFR	1.002 (0.728–1.2750	0.421 (0.130–0.713)	**0.011**
miR-21-5p	Age	0.595 (0.229–0.96)	0.912 (0.558–1.266)	0.251
	HOMA-IR	0.613 (0.222–1.0040	1.029 (0.638–1.420)	0.152
	eGFR	0.655 (0.290–1.021)	0.855 (0.501–1.208)	0.468
miR-132	Age	0.997 (0.620–1.375)	0.818 (0.429–1.208)	0.549
	HOMA-IR	1.134 (0.727–1.541)	0.815 (0.377–1.253)	0.309
	eGFR	1.024 (0.656–1.392)	0.790 (0.410–1.169)	0.413
miR-212-5p	Age	0.415 (0.097–0.733)	0.285 (−0.058–0.627)	0.616
	HOMA-IR	0.353 (0.082–0.625)	0.262 (−0.032–0.556)	0.648
	eGFR	0.414 (0.110–0.718)	0.286 (−0.040–0.612)	0.593

Data are presented as the estimated marginal mean (95% CI) for variables compared with ANCOVA. Statistically significant differences are considered at *p* < 0.05 in bold. Abbreviation: estimated Glomerular Filtration Rate (eGFR); Homeostatic Model Assessment for Insulin Resistance (HOMA-IR).

**Table 4 metabolites-12-00820-t004:** Correlation between miRNAs and symptom questionnaires’ scores and voiding diary parameters.

miRNA	Variables	Correlation Coefficient (r)	*p* Value
miR-592	OABSS	−0.325	**0.047**
	ICIQ-SF	−0.392	**0.016**
	IIQ-7	−0.422	**0.008**
miR-491-5p	OABSS	−0.405	**0.011**
	ICIQ-SF	-0.379	**0.019**
	IIQ-7	−0.494	**0.001**
miR-21-5p	24 h frequency	−0.522	**0.001**
	Daytime frequency	0.45	**0.005**
miR-212-5p	IIQ-7	0.356	**0.026**

The *p* value for Spearman correlation; Statistically significant differences are considered at *p* < 0.05 in bold. Abbreviations: Overactive Bladder Symptom Score (OABSS), International Consultation on Incontinence Questionnaire-Short Form (ICIQ-SF); Incontinence Impact Questionnaire (IIQ-7).

## Data Availability

Data is contained within the article or Appendix A.

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
