# Peer review of "Urinary Levels of miR-491-5p and miR-592 as Potential Diagnostic Biomarkers in Female Aging Patients with OAB: A Pilot Study"

_metabolites, 2022, doi:10.3390/metabo12090820_

Round 1

Reviewer 1 Report

the authors reported their experience on Urinary levels of miR-491-5p and miR-592 as potential diagnostic biomarkers in female aging patients with OAB.

the research was clearly presented.

I would suggest some revisions in order to improve the quality of the paper:

- were patients consecutive?

- please add a flow chart for included and excluded patients

- please add a sample size calculation

- please add a matching pair analysis

Reviewer 2 Report

The paper presented for review concerns the potential use of miR-491-5p and miR-592 levels as a biomarker of the overactive bladder syndrome. The authors point to increased levels of microRNAs in patients compared to controls. The reviewer was interested in the issue of patients suffering from OBA. The authors indicated that samples from patients with and without treatment were selected for the study. The reviewer asks for an explanation of the differences in the levels of the examined indicators in both groups. Did the treatment significantly affect the obtained results? Have the authors considered enlarging the group of patients and treating them separately - those with and without treatment? The treatment used can significantly affect the results. Apart from the indicated doubts, the work is very interesting.

Reviewer 3 Report

The definition of OAB should be based on the ICS glossary.

Most references should be updated.

 What were your selection criteria for OAB? Please mention it in the method section.

 Why did you consider only women aged 50-80 years? Please describe the rationale of your study in the introduction.

What was the study design? How did you match the case and controls, and with which criteria?

All of the abbreviations should be described in the legends of tables.  

Round 2

Reviewer 3 Report

All of my comments responded satisfactorily.